# Near-Optimal Lower Bounds For Convex Optimization For All Orders of Smoothness

**Ankit Garg**
Microsoft Research India
Bengaluru, KA, India
garga@microsoft.com

**Robin Kothari**
Microsoft Quantum and Microsoft Research
Redmond, WA, USA
robin.kothari@microsoft.com

**Praneeth Netrapalli**
Google Research India
Bengaluru, KA, India
praneethn@gmail.com

**Suhail Sherif**
Vector Institute
Toronto, ON, Canada
suhail.sherif@gmail.com

## Abstract

We study the complexity of optimizing highly smooth convex functions. For a positive integer $p$, we want to find an $\epsilon$-approximate minimum of a convex function $f$, given oracle access to the function and its first $p$ derivatives, assuming that the $p$th derivative of $f$ is Lipschitz.

Recently, three independent research groups (Jiang et al., PLMR 2019; Gasnikov et al., PLMR 2019; Bubeck et al., PLMR 2019) developed a new algorithm that solves this problem with $\widetilde{O}(1/\epsilon^{\frac{2}{3p+1}})$ oracle calls for constant $p$. This is known to be optimal (up to log factors) for deterministic algorithms, but known lower bounds for randomized algorithms do not match this bound. We prove a new lower bound that matches this bound (up to log factors), and holds not only for randomized algorithms, but also quantum algorithms.

## 1 Introduction

In recent years, several optimization algorithms have been proposed, especially for machine learning problems, that achieve improved performance by exploiting the smoothness of the function to be optimized. Specifically, these algorithms have better performance when the function's first, second, or higher order derivatives are Lipschitz [Nes08, Bae09, MS13, Nes19, GDG+19, JWZ19, BJL+19b].

In this paper we study the problem of minimizing a highly smooth convex function given black-box access to the function and its higher order derivatives. The simplest example of the family of problems we consider here is the problem of approximately minimizing a convex function $f : \mathbb{R}^n \to \mathbb{R}$, given access to an oracle that on input $x \in \mathbb{R}^n$ outputs $(f(x), \nabla f(x))$, under the assumption that the function's first derivative, its gradient $\nabla f$, has bounded Lipschitz constant. This problem can be solved by Nesterov's accelerated gradient descent, and it is known that this algorithm is optimal (in high dimension) among deterministic and randomized algorithms [Nes83, NY83].

More generally, for any positive integer $p$, consider the $p$th-order optimization problem: For known $R > 0, \epsilon > 0$, and $L_p > 0$, we have a $p$ times differentiable convex function $f : \mathbb{R}^n \to \mathbb{R}$ whose $p$th derivative has Lipschitz constant at most $L_p$, which means

$$\|\nabla^p f(x) - \nabla^p f(y)\| \le L_p \|x - y\|, \tag{1}$$

35th Conference on Neural Information Processing Systems (NeurIPS 2021).

where $\|\cdot\|$ is the $\ell_2$ norm (for vectors) or induced $\ell_2$ norm (for operators). Our goal is to find an $\epsilon$-approximate minimum of this function in a ball of radius $R$, which is any $x^*$ that satisfies

$$f(x^*) - \min_{x \in B_R(0)} f(x) \leq \epsilon, \tag{2}$$

where $B_R(0)$ is the $\ell_2$-ball of radius $R$ around the origin. We can access the function $f$ through a $p$th order oracle, which when queried with a point $x \in \mathbb{R}^n$ outputs

$$(f(x), \nabla f(x), \ldots, \nabla^p f(x)). \tag{3}$$

As usual, $\nabla^p f(x)$ denotes the $p$th derivative of $f(x)$.

Our primary object of study will be the minimum query cost of an algorithm that solves the problem, i.e. the number of queries (or calls) to the oracle in eq. (3) that an algorithm has to make.[1] For a fixed $p$, it seems like this problem has 4 independent parameters, $n$, $L_p$, $R$, and $\epsilon$, but the parameters are not all independent since we can scale the input and output spaces of the function to affect the latter 3 parameters. Thus the complexity of any algorithm can be written as a function of $n$ and $L_p R^{p+1}/\epsilon$. In this paper we focus on the high-dimensional setting where $n$ may be much larger than the other parameters, and the best algorithms in this regime have complexity that only depends on $L_p R^{p+1}/\epsilon$ with no dependence on $n$.

As noted, the $p = 1$ problem has been studied since the early 80s [Nes83, NY83], and the $p > 1$ problem has also been considered [Nes08, MS13]. In an exciting recent development, new algorithms were proposed for all $p$ (with very similar complexity) by three independent groups of researchers: Gasnikov, Dvurechensky, Gorbunov, Vorontsova, Selikhanovych, and Uribe [GDG+19]; Jiang, Wang, and Zhang [JWZ19]; Bubeck, Jiang, Lee, Li and Sidford [BJL+19b]. All three groups develop deterministic algorithms that make

$$\tilde{O}_p \left( \left( L_p R^{p+1}/\epsilon \right)^{2/(3p+1)} \right) \tag{4}$$

oracle calls,[2] where the subscript $p$ in the big Oh (or big Omega) notation means the constant in the big Oh can depend on $p$. In other words, this notation means that we treat $p$ as a constant. This improved upon the bounds of [Bae09, Nes19]; both the works develop deterministic algorithms that make

$$\tilde{O}_p \left( \left( L_p R^{p+1}/\epsilon \right)^{1/(p+1)} \right) \tag{5}$$

oracle calls.

This algorithm is nearly optimal among deterministic algorithms, since the works [Nes19, ASS19] showed that any deterministic algorithm that solves this problem must make $\Omega_p \left( \left( L_p R^{p+1}/\epsilon \right)^{2/(3p+1)} \right)$ queries. However, for randomized algorithms, the known lower bound is weaker. Agarwal and Hazan [AH18] showed that any randomized algorithm must make

$$\Omega_p \left( \left( L_p R^{p+1}/\epsilon \right)^{2/(5p+1)} \right) \tag{6}$$

queries. To the best of our knowledge, no lower bounds are known in the setting of high-dimensional smooth convex optimization against quantum algorithms, although quantum lower bounds are known in the low-dimensional setting [CCLW20, vAGGdW20] and for non-smooth convex optimization [GKNS21].

In this work, we close the gap (up to log factors) between the known algorithm and randomized lower bound for all $p$. Furthermore, our lower bound also holds against quantum algorithms.

**Theorem 1.** *Fix any $p \in \mathbb{N}$. For all $\epsilon > 0$, $R > 0$, $L_p > 0$, there exists an $n > 0$ and a set of $n$-dimensional functions $\mathcal{F}$ with $p$th-order Lipschitz constant $L_p$ (i.e., satisfying eq. (1)) such that any randomized or quantum algorithm that outputs an $\epsilon$-approximate minimum (satisfying eq. (2)) for any function $f \in \mathcal{F}$ must make*

$$\Omega_p \left( \left( L_p R^{p+1}/\epsilon \right)^{2/(3p+1)} \left( \log L_p R^{p+1}/\epsilon \right)^{-2/3} \right) \tag{7}$$

*queries to a $p$th order oracle for $f$ (as in eq. (3)).*

---

[1]For simplicity we assume that the oracle's output is computed to arbitrarily many bits of precision. This only makes our results stronger, since we prove lower bounds in this paper.

[2]Note that the query complexity does not have any dependence on the dimension $n$. Of course, actually implementing each query will take poly$(n)$ time, but we only count the number of queries here.

In fact, this lower bound holds even against highly parallel randomized algorithms, where the algorithm can make $\text{poly}(n, L_p R^{p+1}/\epsilon)$ queries in each round and we only count the total number of query rounds (and not the total number of queries). See [BJL$^+$19a] for previous work in this setting, including speedups for first-order convex optimization in the low dimensional setting.

In this introduction, we have deliberately avoided explaining the quantum model of computation to make the results accessible to readers without a background in quantum computing. The entire paper is written so that the randomized lower bound is fully accessible to any reader who does not wish to understand the quantum model and quantum lower bound. For readers familiar with quantum computing, we note that the only thing to be changed to get the quantum model is to modify the oracle in eq. (3) to support queries in quantum superposition. This is done in the usual way, by defining a unitary implementation of the oracle, which allows quantum algorithms to make superposition queries and potentially solve the problem more efficiently than randomized algorithms.

## 2   High level overview

Let us first consider the lower bound against randomized algorithms. Let us also first look at the special setting of $p = 0$ where we still assume access to the gradient (or $p = 1$) oracle. To be more precise, the oracle returns subgradients, since gradients need not be defined at all points for Lipschitz convex functions. For this setting, known popularly as *nonsmooth* convex optimization, the optimal lower bound of $\Omega\left(\frac{1}{\epsilon^2}\right)$ is in fact a classical result [NY83]. The proof of this result is very elegant and has been used subsequently to prove several other related lower bounds such as for parallel randomized algorithms [Nem94], quantum algorithms [GKNS21] etc. Since our proof builds on this framework, we now review this.

**Nonsmooth lower bound instance.** The lower bound instance for nonsmooth convex optimization is $\min_{\|x\| \leq 1} f_V(x)$ where $f_V : \mathbb{R}^n \to \mathbb{R}$ is chosen as

$$f_V(x) = \max_{i \in [k]} \langle v_i, x \rangle + (k - i)\gamma, \tag{8}$$

with $k = O\left(n^{1/3}\right)$, $\gamma = \tilde{\Theta}\left(\frac{1}{\sqrt{n}}\right)$, and $V = (v_1, \cdots, v_k)$ comprises $k$ orthonormal vectors chosen uniformly at random. The argument essentially shows that

(i) in order to find a $O\left(\frac{1}{\sqrt{k}}\right)$-approximate minimizer, one needs to know all the $v_i$'s, and

(ii) with high probability, each query reveals at most one new vector $v_i$.

This yields a lower bound of $k$ queries for achieving an error of $O\left(\frac{1}{\sqrt{k}}\right)$. Since $f$ is a $1$-Lipschitz function, when we rewrite this bound in terms of error, it yields the $\Omega\left(\frac{1}{\epsilon^2}\right)$ randomized lower bound.

For the $p = 1$ setting, known popularly as *smooth* convex optimization, the optimal lower bound $\Omega\left(\sqrt{\frac{1}{\epsilon}}\right)$ is also a classical result originally proven in [NY83]. However, the proof of this result in [NY83] is quite complicated and is not widely known. The recent papers of [GN15, DG20] provide a much simpler proof of the $p = 1$ result by using the lower bound construction for $p = 0$ setting described above and using *smoothing*, which we now review.

**Smoothing.** Smoothing refers to the process of approximating a given Lipschitz function $f$ by another function $g$, which has Lipschitz continuous $p^{\text{th}}$ derivatives for some $p \geq 1$. Further, since we will be applying this operation to (8), we will describe smoothing in this context.

**Definition 1.** *An operator $\mathcal{S}$ which takes a $1$-Lipschitz function $f$ to another convex function $\mathcal{S}[f]$ is called a $(p, \beta, \epsilon)$-smoothing operation if it satisfies the following:*

1. *Smoothness: $p^{\text{th}}$ derivatives of $\mathcal{S}[f]$ are Lipschitz continuous with parameter $\beta$, and*

2. *Approximation: For any $x$, we have $|f(x) - \mathcal{S}[f](x)| \leq \epsilon$.*

If we can design a smoothing operation as per the above definition with $\epsilon = O\left(1/\sqrt{k}\right)$ and further ensure that property (ii) above i.e., *with high probability, each query to the first $p$ derivatives of $\mathcal{S}[f]$*

*reveals at most one new vector* $v_i$, then the proof strategy of lower bound for nonsmooth convex optimization can be executed on the smoothed instance $\mathcal{S}[f]$, there by giving us a lower bound for $p^{\text{th}}$-order smooth convex optimization. This is the key idea of [GN15, DG20]. Further, the smaller $\beta$ is, the better the bound we obtain. However, since $f$ can have discontinuous $p^{\text{th}}$ derivatives, there is a tension between the approximation property which tries to keep $\mathcal{S}[f]$ close to $f$ and the smoothness property. So, one cannot make $\beta$ very small after fixing $\epsilon = O(1/\sqrt{k})$. For the rest of this section, we fix $\epsilon = O(1/\sqrt{k})$ in Definition 1.

For the $p = 1$ setting, there is a well-known smoothing operation known as *Moreau/inf-conv* smoothing [BC11], which obtains the best possible smoothing with $\beta = \Theta(k^{1.5})$. This gives the tight query lower bound of $\Omega\left(\frac{1}{\sqrt{\epsilon}}\right)$ for smooth convex optimization.

However, there is no known generalization of inf-conv smoothing for $p \geq 2$, so one needs to use a different smoothing operator to extend this proof strategy for proving query lower bounds for higher order smooth convex optimization. Given any $p \geq 1$, [AH18] indeed construct such a smoothing, called *randomized smoothing* which maps Lipschitz convex functions to convex functions with Lipschitz $p^{\text{th}}$ derivatives. In the general $p \geq 1$ setting, a smoothing operator with $\beta = O\left(k^{3p/2}\right)$ would give the optimal lower bound of $\Omega\left(\epsilon^{\frac{-2}{3p+1}}\right)$. However, the randomized smoothing of [AH18] can obtain only $\beta = O\left(k^{5p/2}\right)$ leading to a suboptimal $\Omega\left(\epsilon^{\frac{-2}{5p+1}}\right)$ lower bound for $p^{\text{th}}$ order smooth convex optimization.

We design an improved smoothing operation, for the specific class of functions in Equation (8), with the optimal $\beta = O\left(k^{3p/2}\right)$ using two key ideas. The first idea is the *softmax* function with parameter $\rho$ defined as $\mathsf{smax}_\rho(z) := \rho \log\left(\sum_{i \in [k]} \exp\left(\frac{z_i}{\rho}\right)\right)$, where $z \in \mathbb{R}^k$. If we apply $\mathsf{smax}_\rho$, with $\rho = k^{-3/2}$, to functions of the form (8) through:

$$\mathsf{smax}_\rho(\mathsf{vec}_V(x)) := \rho \log\left(\sum_{i \in [k]} \exp\left(\frac{\langle v_i, x \rangle + (k-i)\gamma}{\rho}\right)\right), \tag{9}$$

where $\mathsf{vec}_V(x) := (\langle v_1, x \rangle + (k-1)\gamma, \langle v_2, x \rangle + (k-2)\gamma, \ldots, \langle v_k, x \rangle)$, we can show that $\mathsf{smax}_\rho(\mathsf{vec}_V(x))$ satisfies Definition 1 with the optimal value of $\beta = O\left(k^{3p/2}\right)$. However, any query on derivatives of $\mathsf{smax}_\rho(\mathsf{vec}_V(x))$ reveals information about all the vectors $v_i$ simultaneously since for instance the gradient is given by

$$\nabla \mathsf{smax}_\rho(\mathsf{vec}_V(x)) = \sum_{i \in [k]} \frac{\exp\left(\frac{\langle v_i, x \rangle + (k-i)\gamma}{\rho}\right)}{\sum_{j \in [k]} \exp\left(\frac{\langle v_j, x \rangle + (k-j)\gamma}{\rho}\right)} \cdot v_i. \tag{10}$$

Consequently, it cannot be directly used to obtain a lower bound. The second idea is that even though the function value and derivatives of $\mathsf{smax}_\rho(\mathsf{vec}_V(x))$ have contribution from all $v_i$'s, the contribution is heavily dominated (i.e., up to $\frac{1}{\mathrm{poly}(k)}$ error) by $v_{i^*(x)}$, where $i^*(x) = \mathrm{argmax}_{i \in [k]} \langle v_i, x \rangle + (k-i)\gamma$, whenever $\langle v_{i^*(x)}, x \rangle + (k - i^*(x))\gamma > \langle v_i, x \rangle + (k-i)\gamma + \Omega(\rho \log k)$ for every $i \neq i^*(x)$.

Based on this insight, we design a new 1-Lipschitz convex function given by $h(x) := \max_{i \in [k]} f_i(x)$ where $f_i(x) := \mathsf{smax}_\rho^{\leq i}(\mathsf{vec}_V(x)) + \rho(k-i)n^{-\alpha}$ for an appropriate $\alpha$ to be chosen later, where $\mathsf{smax}_\rho^{\leq i}(\mathsf{vec}_V(x)) := \rho \log\left(\sum_{j \in [i]} \exp\left(\frac{\langle v_j, x \rangle + (k-j)\gamma}{\rho}\right)\right)$. The key property satisfied by $f_i$ is that $f_i \approx f_j$ implies that $\nabla f_i \approx \nabla f_j$ for any $i, j$. This implies that near points of discontinuous gradients for $h$ i.e., points where $\mathrm{argmax}_{i \in [k]} f_i(x)$ changes, the resulting discontinuity in $\nabla h(x)$ is $O\left(\frac{1}{\mathrm{poly}(k)}\right)$. In contrast, the change in gradients of the original instance $f_V(x)$ near points of discontinuity is $\Omega(1)$. If we apply randomized smoothing to $h$, the resulting function can then be shown to have $p^{\text{th}}$ order Lipschitz constant $\widetilde{O}\left(k^{3p/2}\right)$. The precise details, proved in Lemma 4, are technical and form the bulk of this paper. The same proof strategy immediately yields the same bound on the number of *rounds* for *parallel* randomized algorithms as long as the number of queries in each round is at most $\mathrm{poly}(k)$. The reason is that $\mathrm{poly}(k)$ queries are still not sufficient to obtain information about more than one vector per round. Finally, the same proof strategy can be adapted to the quantum setting using the *hybrid argument* [BBBV97]. See Appendix C for more details.

# 3 Smoothing preliminaries

In this section we look at some smoothing functions and their properties. The proofs of these properties can be found in Appendix A.

Let $B_\eta(x) = \{y \in \mathbb{R}^n : \|y - x\| \leq \eta\}$ be the ball of radius $\eta$ around $x \in \mathbb{R}^n$.

**Definition 2** (Randomized smoothing). *For any function $f : \mathbb{R}^n \to \mathbb{R}$ and real-valued $\eta > 0$, the randomized smoothing operator $S_\eta$ produces a new function $S_\eta[f] : \mathbb{R}^n \to \mathbb{R}$ from $f$ with the same domain and range, defined as*

$$S_\eta[f](x) = \mathop{\mathbb{E}}_{y \in B_\eta(x)}[f(y)]. \tag{11}$$

This smoothing turns non-smooth functions into smooth functions. If we start with a function $f$ that is Lipschitz, then after randomized smoothing, the resulting function's first derivative will be defined and Lipschitz [AH18]. Since we want to construct functions with $p$ derivatives, we define a $p$-fold version of randomized smoothing. Recall that $p$ is the same $p$ as in the introduction (i.e., we are proving lower bounds on the $p$th order optimization problem). This operation also depends on a parameter $\beta$ that we will fix later.

**Definition 3** (Smoothing). *The smoothing operator $\mathcal{S}$ on input $f : \mathbb{R}^n \to \mathbb{R}$ outputs the function*

$$\mathcal{S}[f] = S_{\beta/2^p}[S_{\beta/2^{p-1}}[\cdots S_{\beta/2^2}[S_{\beta/2^1}[f]] \cdots]]. \tag{12}$$

The main properties we require from this smoothing are as follows.

**Lemma 1.** *For the smoothing operator $\mathcal{S}$ defined above, the following statements hold true.*

1. *For any functions $f, g : \mathbb{R}^n \to \mathbb{R}$ for which $\mathcal{S}[f], \mathcal{S}[g]$ are well-defined, $\mathcal{S}[f + g] = \mathcal{S}[f] + \mathcal{S}[g]$.*

2. *The value $\mathcal{S}[f](x)$ only depends on the values of $f$ within a $(1 - 2^{-p})\beta$ radius of $x$.*

3. *The gradient and higher order derivatives of $\mathcal{S}[f]$ at $x$ depend only on the values of $f$ within $B_\beta(x)$.*

4. *If $\nabla^p f$ is $L$-Lipschitz in a ball of radius $\beta$ around $x$, then $\nabla^p \mathcal{S}[f]$ is also $L$-Lipschitz at $x$.*

5. *Let $f$ be $G$-Lipschitz in a ball of radius $\beta$ around $x$. Then $\mathcal{S}[f]$ is $p$-times differentiable, and for any $i \leq p$, $\nabla^i \mathcal{S}[f]$ is $L$-Lipschitz in a $\beta/2^p$-ball around $x$ with $L \leq \frac{n^i 2^{i(i+1)/2}}{\beta^i} G$.*

6. *Let $f$ be $G$-Lipschitz in a ball of radius $\beta$ around $x$. Then $|\mathcal{S}[f](x) - f(x)| \leq \beta G$.*

7. *If $f$ is a convex function, then $\mathcal{S}[f]$ is also a convex function.*

We also use the softmax function introduced earlier.

**Definition 4** (Softmax). *For a real number $\rho$, the softmax function $\mathsf{smax}_\rho : \mathbb{R}^n \to \mathbb{R}$ is defined as*

$$\mathsf{smax}_\rho(x) = \rho \ln\Big( \sum_{i \in [n]} \exp(x_i/\rho) \Big). \tag{13}$$

*Let us also define, for $m \leq n$, $\mathsf{smax}_\rho^{\leq m} : \mathbb{R}^n \to \mathbb{R}$ as*

$$\mathsf{smax}_\rho^{\leq m}(x) = \mathsf{smax}_\rho(x_{\leq m}), \text{ or equivalently, } \mathsf{smax}_\rho^{\leq m}(x) = \rho \ln\Big( \sum_{i \in [m]} \exp(x_i/\rho) \Big). \tag{14}$$

We note the following smoothness properties of softmax.

**Lemma 2.** *The following are true of the function $\mathsf{smax}_\rho$ for any $\rho > 0$.*

1. *The first derivative of $\mathsf{smax}$ can be computed as*

$$\frac{\partial \mathsf{smax}_\rho(x)}{\partial x_i} = \frac{\exp(x_i/\rho)}{\sum_i \exp(x_i/\rho)}. \tag{15}$$

2. $\mathsf{smax}_\rho$ *is 1-Lipschitz and convex.*

3. *The higher-order derivatives of* $\mathsf{smax}$ *satisfy*

$$\|\nabla^p \mathsf{smax}_\rho(x) - \nabla^p \mathsf{smax}_\rho(y)\| \leq \frac{\left(\frac{p+1}{\ln(p+2)}\right)^{p+1} p!}{\rho^p} \|x - y\|. \tag{16}$$

We will also need the following lemma which roughly states that if $\mathsf{smax}(x)$ and $\mathsf{smax}^{\leq m}(x)$ are nearly the same, then their gradients are also nearly the same at $x$.

**Lemma 3.** *Let* $x \in \mathbb{R}^n$ *and* $m < n$. *If*

$$\frac{\mathsf{smax}_\rho(x) - \mathsf{smax}_\rho^{\leq m}(x)}{\rho} = \delta < 1, \tag{17}$$

*Then*

$$\|\nabla \mathsf{smax}_\rho(x) - \nabla \mathsf{smax}_\rho^{\leq m}(x)\| \leq 4\delta. \tag{18}$$

## 4 Function construction and properties

In this section we define the class of functions used in our randomized (and quantum) lower bound, and state the properties of the function that will be exploited in the lower bound. Some proofs have been moved to Appendix B due to space constraints.

Let $k \in \mathbb{N}, \gamma, \beta, \rho, \alpha \in \mathbb{R}$ be parameters to be defined shortly. $k$ and $\gamma$ are parameters as used in the high level overview (Equation (8)), $\beta$ is the parameter required to define $\mathcal{S}$ and $\rho$ is the parameter used in the definition of $\mathsf{smax}$.

**Function construction.** Given a list of orthonormal vectors $v_1, \ldots, v_k \in \mathbb{R}^n$, which we collectively call $V$, we recall that $\mathsf{vec}_V(x) \in \mathbb{R}^k$ denotes the vector

$$\mathsf{vec}_V(x) = (\langle v_1, x \rangle + (k-1)\gamma, \langle v_2, x \rangle + (k-2)\gamma, \ldots, \langle v_k, x \rangle). \tag{19}$$

We can now define our hard function class as follows.

**Definition 5.** *Let* $V = (v_1, \ldots, v_k) \in \mathbb{R}^n$ *be a set of orthonormal vectors. The functions* $f_1, \ldots, f_k$, $h$ *and* $g$ *depend on* $V$ *as follows. Define, for each* $i \in [k]$, *the function* $f_i : \mathbb{R}^n \to \mathbb{R}$ *as*

$$f_i(x) = \mathsf{smax}_\rho^{\leq i}(\mathsf{vec}_V(x)) + \rho(k-i)n^{-\alpha}. \tag{20}$$

*Define* $h(x) = \max_{i \in [k]} f_i(x)$, *and* $g(x) = \mathcal{S}[h](x)$.

Note that in the above definition we apply the $\mathsf{smax}$ functions on $\mathsf{vec}_V(x)$ and not on $x$. However, since $\mathsf{vec}_V(x)$ is obtained by applying a unitary transform on $x$ and then translating it, the observations about $\mathsf{smax}(x)$ in Lemmas 2 and 3 also hold for $\mathsf{smax}(\mathsf{vec}_V(x))$.

We set $\gamma = 40\sqrt{\frac{\ln n}{n}}$, $k = \lfloor (0.1/\gamma)^{2/3} \rfloor$ (or $\gamma \approx 0.1/k\sqrt{k}$), $\rho = \gamma/100\alpha \ln n$, $\beta = \gamma/\ln n$, $\alpha = p + 1$.

**Function properties.** We now state some properties of the function that will be used to show the lower bounds.

**Lemma 4.** *For any choice of* $V$, *the function* $g$ *is convex, p-times differentiable and satisfies*

$$\|\nabla^p g(x) - \nabla^p g(y)\| \leq L_p \|x - y\| \tag{21}$$

*where* $L_p \leq O_p(k^{3p/2}(\ln k)^p)$.

The proof relies on the fact that $\mathsf{smax}$ is smooth and hence each $f_i$ is smooth. If $h = f_i$ for a particular $i$ in a $\beta$-neighborhood of $x$, then $g = \mathcal{S}[h]$ would also be smooth (by Lemma 1, item 4). If $h$ depends on multiple $f_i$s in a $\beta$-neighborhood of $x$, then we know that at least two softmax's involved in the definitions of the $f_i$s have nearly the same value in the neighborhood of $x$, and by Lemma 3 they have nearly the same gradient. This makes $h$ *nearly* smooth, which will allow us to say that $g$ is smooth at $x$ (by Lemma 1, item 5).

*Proof.* Each $f_i$ is an instance of softmax applied to $\text{vec}_V(x)$ plus a constant. Since $\text{vec}_V(x)$ is the vector $x$ transformed by a unitary and then translated, the smoothness and convexity properties of $\text{smax}_\rho$ also apply to $f_i$. Hence each $f_i$ is convex, $p$-times differentiable and its $p$th derivatives are $O_p(\rho^{-p})$-Lipschitz (see Lemma 2). The function $h$, being a maximum over convex functions, is also convex. By the properties of the smoothing operator $\mathcal{S}$ (Lemma 1), the function $g = \mathcal{S}[h]$ is also convex.

Let $x \in \mathbb{R}^n$. Let $j \in [k]$ be the minimum number such that there is a point $y \in B_\beta(x)$ for which $h(y) = f_j(y)$. We can rewrite $h$ as follows: $h(z) = f_j(z) + \max_{i>j}(f_i(z) - f_j(z))$. We call $f_j(z)$ the smooth term and $\max_{i>j}(f_i(z) - f_j(z))$ the non-smooth term. We know that $f_j$ has an $O_p(\rho^{-p})$ upper bound on the Lipschitzness of its $p$-th order derivatives. If all points $y \in B_\beta(x)$ satisfy $h(y) = f_j(y)$, then the non-smooth term is $0$ and so it does not change the smoothness of $h$. $g = \mathcal{S}[h]$ will maintain this smoothness (see item 4 of Lemma 1).

If the non-smooth term is non-zero at some point in $B_\beta(x)$, then we wish to show that the non-smooth term has a small Lipschitz constant in $B_\beta(x)$. This would imply, via item 5 of Lemma 1, that the $p$-th order derivative of the smoothing of the non-smooth term with $\mathcal{S}$ would have a small Lipschitz constant. Towards this let $x'$ be any point in $B_\beta(x)$. Let $I_{x'}$ be the set $\{i \in [k] | h(x') = f_i(x')\}$. The set of subgradients of the non-smooth term at $x'$ is the convex hull of $\{\nabla(f_i - f_j)(x')\}_{i \in I(x')}$. So if we show that for an arbitrary $i \in I(x')$, $\|\nabla(f_i - f_j)(x')\| \leq L$, then we know that the non-smooth part is $L$-Lipschitz at $x'$. If $i = j$, then the gradient is zero. Let us take an $i \neq j$ (since $j$ is the smallest, in fact $i > j$). By convexity of the ball and the continuity of $f_i$ and $f_j$, there must be a point $y$ in $B_\beta(x)$ for which $h(y) = f_i(y) = f_j(y)$. Note that $x' \in B_{2\beta}(y)$.

The statement $f_i(y) = f_j(y)$ translates to

$$\frac{\text{smax}_\rho^{\leq i}(\text{vec}_V(y)) - \text{smax}_\rho^{\leq j}(\text{vec}_V(y))}{\rho} = (i - j)n^{-\alpha} << 1. \tag{22}$$

Expanding the expression for $\text{smax}_\rho^{\leq i}$ and $\text{smax}_\rho^{\leq j}$ we get

$$\frac{\text{smax}_\rho^{\leq i}(\text{vec}_V(y)) - \text{smax}_\rho^{\leq j}(\text{vec}_V(y))}{\rho} = \ln\left(\frac{\sum_{\ell=1}^{i} \exp\left(\frac{\langle y, v_\ell\rangle + (k-\ell)\gamma}{\rho}\right)}{\sum_{\ell=1}^{j} \exp\left(\frac{\langle y, v_\ell\rangle + (k-\ell)\gamma}{\rho}\right)}\right) \tag{23}$$

$$= \ln\left(1 + \frac{\sum_{\ell=j+1}^{i} \exp\left(\frac{\langle y, v_\ell\rangle + (k-\ell)\gamma}{\rho}\right)}{\sum_{\ell=1}^{j} \exp\left(\frac{\langle y, v_\ell\rangle + (k-\ell)\gamma}{\rho}\right)}\right) \tag{24}$$

Since $\|x' - y\| \leq 2\beta$, we have that $|\langle x', v\rangle - \langle y, v\rangle| \leq 2\beta$ for any unit vector $v$. Hence

$$\frac{\text{smax}_\rho^{\leq i}(\text{vec}_V(x')) - \text{smax}_\rho^{\leq j}(\text{vec}_V(x'))}{\rho} \leq \ln\left(1 + \frac{e^{2\beta/\rho}\sum_{\ell=j+1}^{i} \exp\left(\frac{\langle y, v_\ell\rangle + (k-\ell)\gamma}{\rho}\right)}{e^{-2\beta/\rho}\sum_{\ell=1}^{j} \exp\left(\frac{\langle y, v_\ell\rangle + (k-\ell)\gamma}{\rho}\right)}\right). \tag{25}$$

For all $c > 0$, $\ln(1 + e^{4\beta/\rho}c) \leq e^{4\beta/\rho}\ln(1 + c)$. So we can conclude from the above that

$$\frac{\text{smax}_\rho^{\leq i}(\text{vec}_V(x')) - \text{smax}_\rho^{\leq j}(\text{vec}_V(x'))}{\rho} \leq (i - j)n^{-\alpha}e^{4\beta/\rho}. \tag{26}$$

Now by Lemma 3, $\|\nabla(f_i - f_j)(x')\| = \|\nabla\text{smax}_\rho^{\leq i}(\text{vec}_V(x')) - \nabla\text{smax}_\rho^{\leq j}(\text{vec}_V(x'))\| \leq 4(i - j)n^{-\alpha}e^{4\beta/\rho}$.

Hence the non-smooth part of $h$ is $4kn^{-\alpha}\exp(8\beta/\rho)$-Lipschitz in $B_\beta(x)$. The $p$th derivatives of $g = \mathcal{S}[h] = \mathcal{S}[f_j] + \mathcal{S}[\max_{i>j}(f_i - f_j)]$ are thus by Lemma 1, $O_p(\rho^{-p} + n^p\beta^{-p}kn^{-\alpha}\exp(4\beta/\rho))$-Lipschitz. We know that $\alpha = p + 1$, $\beta = \gamma/\ln n$ and $\rho = \gamma/100\alpha \ln n$, simplifying our bound to $O_p((\ln n/\gamma)^p)$. Furthermore, $k = \lfloor (0.1/\gamma)^{2/3}\rfloor$ and $\ln n = O(\ln k)$. Hence we can rewrite our upper bound as $O_p\left(k^{3p/2}(\ln k)^p\right)$. $\qquad\square$

We now see how to prove the query lower bound on optimizing this function class. In order to do so, we need to introduce some intermediate functions. Let $h_i(x) = \max_{j \in [i]} f_i(x)$ and $g_i(x) = \mathcal{S}[h_i](x)$. Let an oracle call to a function $f$ at a point $x$ be denoted by $\mathsf{Q}_f(x) = (f(x), \nabla f(x), \nabla^2 f(x), \ldots, \nabla^p f(x))$. The following results will hold when the set $V$ of orthonormal vectors is chosen uniformly at random (or Haar randomly). The next two lemmas about these intermediate functions form the backbone of our lower bound.

**Lemma 5.** *Fix any $t \in [0, \ldots, k-1]$. Let $V$ be distributed Haar randomly. Conditioned on any fixing of $\{v_i\}_{i \leq t}$, any query $x$ in the unit ball will satisfy $\mathsf{Q}_g(x) = \mathsf{Q}_{g_{t+1}}(x)$ with probability at least $1 - 1/n^{10}$.*

**Lemma 6.** *Let $V$ be distributed Haar randomly. Conditioned on any fixing of $\{v_i\}_{i \leq k-1}$, any point $x$ in the unit ball will be $\epsilon$-optimal for $g$ with probability at most $1/n^{10}$.*

To see how the lemmas above lead us to our lower bound, fix any $k-1$-query algorithm and consider the following experiment. For each $i$ from 1 to $k-1$, when the algorithm makes its $i$th query do the following.

- Sample $v_i$ from the space orthogonal to the vectors $v_1$ to $v_{i-1}$.
- Provide the algorithm the value that $\mathsf{Q}_{g_i}$ returns on the query. Note that the function $g_i$ depends only on the sampled vectors $v_1$ through $v_i$.

It follows that the output of the algorithm is independent of the vector $v_k$ (conditioned on the vectors $v_1$ through $v_{k-1}$). Now we use Lemma 6 to say that with high probability the output of the algorithm is not $\epsilon$-optimal for $g$. We can now use Lemma 5 along with the hybrid argument to conclude that with high probability the transcript of this query algorithm is the same as the actual transcript (i.e. the transcript had $v_1$ to $v_k$ all been sampled at the beginning and all the queries been to $\mathsf{Q}_g$). Since the transcripts are the same with high probability, the outputs of the algorithms are also the same with high probability. Hence even when all the queries are to $\mathsf{Q}_g$, with high probability the output is not $\epsilon$-optimal for $g$. This proof is made formal as the proof of Theorem 2.

## 5   Lower bounds

We can now establish the randomized lower bound using Lemma 4, Lemma 5, and Lemma 6.

**Theorem 2.** *Let $\mathcal{A}$ be a randomized query algorithm making at most $k-1$ queries to $\mathsf{Q}_g$. When $V$ is distributed Haar randomly, the probability that the output of $\mathcal{A}$ is $\epsilon$-optimal is $o(1)$.*

*Proof.* Let the success probability of $\mathcal{A}$ be $p_{\text{succ}}$ when $V$ is distributed Haar randomly. We can fix the randomness of $\mathcal{A}$ to get a deterministic algorithm $\mathcal{B}$ with success probability at least $p_{\text{succ}}$ on the same distribution.

Let us denote the transcript of $\mathcal{B}$ as $\overline{x} = (x_1, x_2, \ldots, x_{k-1}, x_{\text{out}})$ where $x_i$ is the $i$th query made and $x_{\text{out}}$ is the output of the algorithm. Note that these are random variables that depend only on $V$. We now create hybrid transcripts $\overline{x}^{(i)}$ for $0 \leq i \leq k-1$. The hybrid transcript $\overline{x}^{(i)} = (x_1^{(i)}, \cdots, x_{k-1}^{(i)}, x_{\text{out}}^{(i)})$ is defined as the transcript of $\mathcal{B}$ when, for all $j \leq i$, its $j$th oracle call (which is supposed to be to $\mathsf{Q}_g$) is replaced with an oracle call to $\mathsf{Q}_{g_j}$. Note that

- For any $V$, $\overline{x} = \overline{x}^{(0)}$.

- $\overline{x}^{(k-1)}$ is a function of $\{v_i\}_{i \leq k-1}$.

- For any $V$, if $\mathsf{Q}_g(x_i^{(i-1)}) = \mathsf{Q}_{g_i}(x_i^{(i)})$ then $\overline{x}^{(i-1)} = \overline{x}^{(i)}$. This is because they have queried the same oracles in their first $i-1$ calls, given the same input in the $i$th call and gotten the same output, and have been querying the same oracles thereafter.

We start with the observation that

$$\Pr_V[x_{\text{out}}^{(k-1)} \text{ is } \epsilon\text{-optimal}] = \mathop{\mathbb{E}}_{v_1, \ldots, v_{k-1}} \left[ \Pr_{v_k | v_{<k}} [x_{\text{out}}^{(k-1)} \text{ is } \epsilon\text{-optimal}] \right] \tag{27}$$

$$\leq n^{-10}. \tag{by Lemma 6}$$

Next we show that $\Pr_V[x_{\text{out}}^{(k-1)} = x_{\text{out}}] \geq 1 - o(1)$ which will complete the proof.

$$\Pr_V[x_{\text{out}} \neq x_{\text{out}}^{(k-1)}] \leq \sum_{i \in [k-1]} \Pr_V[x_{\text{out}}^{(i-1)} \neq x_{\text{out}}^{(i)}] \tag{28}$$

$$\leq \sum_{i \in [k-1]} \Pr_V[\overline{x}^{(i-1)} \neq \overline{x}^{(i)}] \tag{29}$$

$$\leq \sum_{i \in [k-1]} \Pr_V[\mathsf{Q}_g(x_i^{(i-1)}) \neq \mathsf{Q}_{g_i}(x_i^{(i)})] \tag{30}$$

$$\leq \sum_{i \in [k-1]} \mathop{\mathbb{E}}_{v_1,\dots,v_{i-1}} \left[ \Pr_{v_i,\dots,v_k|v_{<i}}[\mathsf{Q}_g(x_i^{(i-1)}) \neq \mathsf{Q}_{g_i}(x_i^{(i)})] \right] \tag{31}$$

$$\leq kn^{-10}, \tag{32}$$

since $x_i^{(i-1)} = x_i^{(i)}$, and using Lemma 5. $\qquad\square$

We now translate the above lower bound to the optimization setting and establish the randomized lower bound in Theorem 1.

*Proof of randomized lower bound in Theorem 1.*
Our hard function class had $L_p = O_p(k^{3p/2}(\ln k)^p)$, $R = 1$, $\epsilon = 0.1/\sqrt{k}$. Given these parameters, $L_p R^{p+1}/\epsilon = O_p(k^{(3p+1)/2}(\ln k)^p)$ and $\ln(L_p R^{p+1}/\epsilon) = O_p(\ln k)$. Hence

$$\left(\frac{L_p R^{p+1}}{\epsilon}\right)^{\frac{2}{3p+1}} \left(\ln \frac{L_p R^{p+1}}{\epsilon}\right)^{-2/3} \leq O_p\left(k(\ln k)^{\frac{2p}{3p+1}}(\ln k)^{-2/3}\right) \leq O_p(k). \tag{33}$$

Since we have a lower bound of $k$, the theorem statement follows. $\qquad\square$

We mention here that Lemmas 5 and 6 have even more far-reaching consequences. In Appendix C, we note that these lemmas also prove a lower bound of $k$ in (a) the parallel randomized setting where polynomially many non-adaptive queries are allowed in each round and we want to bound the number of rounds and (b) the quantum setting. Hence the best known deterministic algorithm that was recently discovered is also nearly optimal amongst parallel randomized and quantum algorithms.

## 6 Conclusion

In this paper, we obtained near optimal oracle lower bounds for $p^{\text{th}}$-order smooth convex optimization, for any constant $p$, for randomized algorithms. Our results further hold for quantum algorithms and parallel randomized algorithms as well. To obtain our results, we introduce a new smoothing operator that could be of independent interest. An interesting open problem is to obtain both tight upper and lower bounds when we have oracle access only up to $q^{\text{th}}$-order derivatives even though the function is guaranteed to have Lipschitz $p^{\text{th}}$-order derivatives, for $q < p$.

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
