# A  Deferred proofs: Smoothing preliminaries

Recall the definitions of our smoothing operators.

Let $B_\eta(x)$ be the ball of radius $\eta$ around $x$. For any function $f : \mathbb{R} \to \mathbb{R}$ and real-valued $\eta > 0$, define the function $S_\eta[f] : \mathbb{R}^n \to \mathbb{R}$ as $S_\eta[f](x) = \mathbb{E}_{y \in B_\eta(x)}[f(y)]$.

**Definition 3** (Smoothing). *The smoothing operator $\mathcal{S}$ on input $f : \mathbb{R}^n \to \mathbb{R}$ outputs the function*

$$\mathcal{S}[f] = S_{\beta/2^p}[S_{\beta/2^{p-1}}[\cdots S_{\beta/2^2}[S_{\beta/2^1}[f]]\cdots]]. \tag{12}$$

We claimed the following about $\mathcal{S}$.

**Lemma 1.** *For the smoothing operator $\mathcal{S}$ defined above, the following statements hold true.*

1. *For any functions $f, g : \mathbb{R}^n \to \mathbb{R}$ for which $\mathcal{S}[f], \mathcal{S}[g]$ are well-defined, $\mathcal{S}[f + g] = \mathcal{S}[f] + \mathcal{S}[g]$.*

2. *The value $\mathcal{S}[f](x)$ only depends on the values of $f$ within a $(1 - 2^{-p})\beta$ radius of $x$.*

3. *The gradient and higher order derivatives of $\mathcal{S}[f]$ at $x$ depend only on the values of $f$ within $B_\beta(x)$.*

4. *If $\nabla^p f$ is $L$-Lipschitz in a ball of radius $\beta$ around $x$, then $\nabla^p \mathcal{S}[f]$ is also $L$-Lipschitz at $x$.*

5. *Let $f$ be $G$-Lipschitz in a ball of radius $\beta$ around $x$. Then $\mathcal{S}[f]$ is $p$-times differentiable, and for any $i \leq p$, $\nabla^i \mathcal{S}[f]$ is $L$-Lipschitz in a $\beta/2^p$-ball around $x$ with $L \leq \frac{n^i 2^{i(i+1)/2}}{\beta^i} G$.*

6. *Let $f$ be $G$-Lipschitz in a ball of radius $\beta$ around $x$. Then $|\mathcal{S}[f](x) - f(x)| \leq \beta G$.*

7. *If $f$ is a convex function, then $\mathcal{S}[f]$ is also a convex function.*

While most of the above statements are proven below from first principles, a couple of them follow from [AH18, Corollary 2.4]. For these statements, we only detail the main ideas involved in their proofs.

*Proof.* We prove the above statements in order.

1. This is a simple consequence of the linearity of expectation.

2. By expanding the expectations in the definition of $\mathcal{S}$, we get that $\mathcal{S}[f](x) = \mathbb{E}_{y \sim \mu_x}[f](x)$ where $\mu_x$ is a distribution supported in $B_{(1-2^{-p})\beta}(x)$.

3. The gradient and higher order derivatives of $\mathcal{S}[f]$ at $x$ depend only on the values of $\mathcal{S}[f]$ in an open ball around $x$, say $B_{2^{-p}\beta}(x)$. For any $y \in B_{2^{-p}\beta}(x)$, $\mathcal{S}[f](y)$ depends only on values of $f$ in $B_{(1-2^{-p})\beta}(y) \subseteq B_\beta(x)$.

4. This follows from the proof of [AH18, Corollary 2.4]. It is easy to see that if $f$ is $L$-Lipschitz then $S_\eta[f]$ is also $L$-Lipschitz. $\nabla f$ being $L$-Lipschitz is equivalent to saying that for any unit vector $v \in \mathbb{R}^n$, $g_v(x) := \nabla f(x)[v]$ is an $L$-Lipschitz function. However by linearity of $S_\eta$, $S_\eta[g_v](x) = \nabla S_\eta[f](x)[v]$. Hence $\nabla S_\eta[f]$ is $L$-Lipschitz. A repeated usage of this argument as done in [AH18] proves the statement.

5. The proof of this is via a repeated usage of [AH18, Lemma 2.3] as done in [AH18, Corollary 2.4]. They argue via Stoke's theorem that $S_\eta[f]$ is differentiable even when $f$ may not be differentiable in a set of measure 0, and that $\nabla S_\eta[f]$ is $\frac{n}{\eta} G$-Lipschitz. They then use directional derivatives to inductively show via the same argument the Lipschitzness of the higher-order derivatives.

6. This is a simple consequence of the fact that $\mathcal{S}[f](x)$ is a convex combination of the values $f(y)$ for $y \in B_\beta(x)$.

7. For $y \in B_\beta(\vec{0})$, let $f_y$ be defined as $f_y(x) = f(x + y)$. Since $f$ is convex, it follows that each $f_y$ is convex. Also $\mathcal{S}[f] = \mathbb{E}_{y \in B_\beta(\vec{0})}[f_y]$. Since $\mathcal{S}[f]$ is a convex combination of convex functions, it follows that $\mathcal{S}[f]$ is convex. $\qquad\square$

Recall the definition of the softmax function.

**Definition 4** (Softmax). *For a real number $\rho$, the softmax function $\mathsf{smax}_\rho : \mathbb{R}^n \to \mathbb{R}$ is defined as*

$$\mathsf{smax}_\rho(x) = \rho \ln\Big( \sum_{i \in [n]} \exp(x_i/\rho) \Big). \tag{13}$$

*Let us also define, for $m \leq n$, $\mathsf{smax}_\rho^{\leq m} : \mathbb{R}^n \to \mathbb{R}$ as*

$$\mathsf{smax}_\rho^{\leq m}(x) = \mathsf{smax}_\rho(x_{\leq m}), \text{ or equivalently, } \mathsf{smax}_\rho^{\leq m}(x) = \rho \ln\Big( \sum_{i \in [m]} \exp(x_i/\rho) \Big). \tag{14}$$

We claimed the following about the softmax function.

**Lemma 2.** *The following are true of the function $\mathsf{smax}_\rho$ for any $\rho > 0$.*

1. *The first derivative of $\mathsf{smax}$ can be computed as*

$$\frac{\partial \mathsf{smax}_\rho(x)}{\partial x_i} = \frac{\exp(x_i/\rho)}{\sum_i \exp(x_i/\rho)}. \tag{15}$$

2. *$\mathsf{smax}_\rho$ is 1-Lipschitz and convex.*

3. *The higher-order derivatives of $\mathsf{smax}$ satisfy*

$$\|\nabla^p \mathsf{smax}_\rho(x) - \nabla^p \mathsf{smax}_\rho(y)\| \leq \frac{\left(\frac{p+1}{\ln(p+2)}\right)^{p+1} p!}{\rho^p} \|x - y\|. \tag{16}$$

*Proof.* We prove the statements in order.

1. This is straightforward.

2. We can conclude the 1-Lipschitzness by looking at the norm of the gradients.

$$\|\nabla \mathsf{smax}_\rho(x)\| = \frac{\|(\exp(x_1/\rho), \ldots, \exp(x_n/\rho))\|}{\sum_i \exp(x_i/\rho)} \leq \frac{\sum_i |\exp(x_i/\rho)|}{\sum_i \exp(x_i/\rho)} = 1. \tag{34}$$

The convexity follows from analyzing the Hessian. We get the following as the Hessian.

$$\nabla^2 \mathsf{smax}_\rho(x)_{i,j} = \begin{cases} \frac{1}{\rho}\left( \frac{\exp(x_i/\rho)}{\sum_{t \in [n]} \exp(x_t/\rho)} - \frac{\exp(2x_i/\rho)}{(\sum_{t \in [n]} \exp(x_t/\rho))^2} \right) & \text{if } i = j \\[2ex] \frac{1}{\rho}\left( -\frac{\exp((x_i + x_j)/\rho)}{(\sum_{t \in [n]} \exp(x_t/\rho))^2} \right) & \text{otherwise} \end{cases} \tag{35}$$

Let $v \in \mathbb{R}^n$ denote the column vector $\nabla \mathsf{smax}_\rho(x)$. Then $\nabla^2 \mathsf{smax}_\rho(x) = \frac{1}{\rho}(\operatorname{diag}(v) - vv^\mathsf{T})$. The convexity of $\mathsf{smax}_\rho$ is equivalent to $\nabla^2 \mathsf{smax}_\rho(x)$ being positive semidefinite for all $x$. Since $\rho > 0$, it suffices to prove that $M = \rho \nabla^2 \mathsf{smax}_\rho(x)$ is positive semidefinite. To this end, let $y \in \mathbb{R}^n$ be any column vector.

$$y^\mathsf{T} M y = \sum_{i \in [n]} y_i^2 v_i - \langle y, v \rangle^2 \tag{36}$$

$$= \Big( \sum_{i \in [n]} y_i^2 v_i \Big)\Big( \sum_{i \in [n]} v_i \Big) - \Big( \sum_{i \in [n]} y_i v_i \Big)^2 \quad (\text{since } \sum_{i \in [n]} v_i = 1)$$

$$\geq 0. \tag{37}$$

The last inequality follows by using the Cauchy-Schwarz inequality on the vectors $(y_i \sqrt{v_i})_{i \in [n]}$ and $(\sqrt{v_i})_{i \in [n]}$. Here we use the fact that each $v_i$ is nonnegative.

3. This is proven in [Bul20, Theorem 7]. □

**Lemma 3.** *Let $x \in \mathbb{R}^n$ and $m < n$. If*

$$\frac{\mathsf{smax}_\rho(x) - \mathsf{smax}_\rho^{\leq m}(x)}{\rho} = \delta < 1, \tag{17}$$

*Then*

$$\|\nabla\mathsf{smax}_\rho(x) - \nabla\mathsf{smax}_\rho^{\leq m}(x)\| \leq 4\delta. \tag{18}$$

*Proof.* $\frac{\mathsf{smax}_\rho(x) - \mathsf{smax}_\rho^{\leq m}(x)}{\rho} = \delta$ implies that

$$\delta = \ln\left(\frac{\sum_{i=1}^n \exp\left(\frac{x_i}{\rho}\right)}{\sum_{i=1}^m \exp\left(\frac{x_i}{\rho}\right)}\right) = \ln\left(1 + \frac{\sum_{i=m+1}^n \exp\left(\frac{x_i}{\rho}\right)}{\sum_{i=1}^m \exp\left(\frac{x_i}{\rho}\right)}\right) \tag{38}$$

Let $c = \frac{\sum_{i=m+1}^n \exp(x_i/\rho)}{\sum_{i=1}^m \exp(x_i/\rho)}$. Since $\delta = \ln(1+c) \geq c/2$ for $\delta < 1$, an upper bound of $2c$ would suffice to prove the lemma.

Using the equation for the gradient of $\mathsf{smax}$ from Lemma 2, we get $\|\nabla\mathsf{smax}_\rho(x) - \nabla\mathsf{smax}_\rho^{\leq m}(x)\|$ to be equal to

$$\frac{\left(\exp\left(\frac{x_1}{\rho}\right),\ldots,\exp\left(\frac{x_n}{\rho}\right)\right)}{\sum_{i=1}^n \exp\left(\frac{x_i}{\rho}\right)} - \frac{\left(\exp\left(\frac{x_1}{\rho}\right),\ldots,\exp\left(\frac{x_m}{\rho}\right),0,\ldots,0\right)}{\sum_{i=1}^m \exp\left(\frac{x_i}{\rho}\right)} \tag{39}$$

$$= \frac{\left(\exp\left(\frac{x_1}{\rho}\right),\ldots,\exp\left(\frac{x_n}{\rho}\right)\right)}{\sum_{i=1}^n \exp\left(\frac{x_n}{\rho}\right)} - \frac{(1+c)\left(\exp\left(\frac{x_1}{\rho}\right),\ldots,\exp\left(\frac{x_m}{\rho}\right),0,\ldots,0\right)}{(1+c)\sum_{i=1}^m \exp\left(\frac{x_i}{\rho}\right)} \tag{40}$$

$$= \frac{-c\left(\exp\left(\frac{x_1}{\rho}\right),\ldots,\exp\left(\frac{x_m}{\rho}\right)\right) + \left(0,\cdots,0,\exp\left(\frac{x_{m+1}}{\rho}\right),\ldots,\exp\left(\frac{x_n}{\rho}\right),0,\ldots,0\right)}{(1+c)\sum_{i=1}^m \exp\left(\frac{x_i}{\rho}\right)} \tag{41}$$

The norm of this is at most

$$\frac{\|-c\left(\exp\left(\frac{x_1}{\rho}\right),\ldots,\exp\left(\frac{x_m}{\rho}\right)\right)\| + \|\left(\exp\left(\frac{x_{m+1}}{\rho}\right),\ldots,\exp\left(\frac{x_n}{\rho}\right)\right)\|}{(1+c)\sum_{i=1}^m \exp\left(\frac{x_i}{\rho}\right)} \tag{42}$$

$$\leq \frac{c}{1+c}\frac{\sum_{i=1}^m \left|\exp\left(\frac{x_i}{\rho}\right)\right|}{\sum_{i=1}^m \exp\left(\frac{x_i}{\rho}\right)} + \frac{1}{1+c}\frac{\sum_{i=m+1}^n \left|\exp\left(\frac{x_i}{\rho}\right)\right|}{\sum_{i=1}^m \exp\left(\frac{x_i}{\rho}\right)} \tag{43}$$

$$\leq \frac{c}{1+c} + \frac{c}{1+c} < 2c. \qquad\qquad\qquad\square$$

# B  Deferred proofs: Function construction and properties

Here we prove Lemmas 5 and 6.

**Lemma 5.** *Fix any $t \in [0,\ldots,k-1]$. Let $V$ be distributed Haar randomly. Conditioned on any fixing of $\{v_i\}_{i\leq t}$, any query $x$ in the unit ball will satisfy $\mathsf{Q}_g(x) = \mathsf{Q}_{g_{t+1}}(x)$ with probability at least $1 - 1/n^{10}$.*

**Lemma 6.** *Let $V$ be distributed Haar randomly. Conditioned on any fixing of $\{v_i\}_{i\leq k-1}$, any point $x$ in the unit ball will be $\epsilon$-optimal for $g$ with probability at most $1/n^{10}$.*

The proofs of both of these use the following lemma.

**Lemma 7.** *Fix any $t \in [0,\ldots,k-1]$. Conditioned on any fixing of $\{v_i\}_{i\leq t}$, any query $x$ in the unit ball will, with probability $1 - 1/n^{10}$, satisfy $\forall i > t \;\; |\langle v_i, x\rangle| \leq 10\sqrt{\frac{\ln n}{n}}$.*

*Proof.* Note that for $i > t$, $v_i$ is distributed uniformly at random from a unit sphere in $\mathbb{R}^{n-t}$. The following useful concentration statement about random unit vectors follows from [Bal97, Lemma 2.2].

**Proposition 3.** *Let $x \in B(\vec{0}, 1)$. Then for a random unit vector $v$, and all $c > 0$,*

$$\Pr_v(|\langle x, v \rangle| \geq c) \leq 2e^{-nc^2/2}. \tag{44}$$

Using Proposition 3 and the fact that $n - t > n/2$ we have that for any $x$ in the unit ball,

$$\Pr\left[|\langle v_i, x \rangle| \geq 10\sqrt{\frac{\ln n}{n}}\right] \leq 2e^{-n/2\left(10\sqrt{\frac{\ln n}{n}}\right)^2/2}$$

$$\leq 2e^{-25\ln n} \leq n^{-24}.$$

Applying a union bound for each of the vectors $v_{t+1}, \ldots, v_k$, we have that with probability at least $1 - 1/n^{23}$, $\forall i > t$ $|\langle v_i, x \rangle| \leq 10\sqrt{\frac{\ln n}{n}}$. (We use the constant 10 in the lemma statement only because it is a nicer constant than 23.) $\qquad\square$

*Proof of Lemma 5.* To show that $g(x) = g_{t+1}(x)$, we will show that for all $y \in B_\beta(x)$, $h(y) = h_{t+1}(y)$. Let $E_x$ be the event that $x$ satisfies $\forall i > t$ $|\langle v_i, x \rangle| \leq 10\sqrt{\frac{\ln n}{n}}$. We will show that $E_x \implies g(x) = g_{t+1}(x)$. Hence let us assume $E_x$ holds.

We know that $x$ satisfies $\langle v_j, x \rangle - \langle v_{t+1}, x \rangle \leq 20\sqrt{\frac{\ln n}{n}}$ for all $j > t + 1$. Hence for any $y \in B_\beta(x)$, $\langle v_j, y \rangle - \langle v_{t+1}, y \rangle \leq 20\sqrt{\frac{\ln n}{n}} + 2\beta$. To show that $h(y) = h_{t+1}(y)$, it is sufficient to show that $f_{t+1}(y) \geq f_j(y)$ for all $j > t + 1$.

Note that $f_{t+1}(y) \geq f_j(y)$ if and only if

$$(j - t - 1)n^{-\alpha} \geq \ln\left(\frac{\sum_{\ell=1}^{j} \exp\left(\frac{\langle y, v_\ell \rangle + (k-\ell)\gamma}{\rho}\right)}{\sum_{\ell=1}^{t+1} \exp\left(\frac{\langle y, v_\ell \rangle + (k-\ell)\gamma}{\rho}\right)}\right)$$

$$= \ln\left(1 + \frac{\sum_{\ell=t+2}^{j} \exp\left(\frac{\langle y, v_\ell \rangle + (k-\ell)\gamma}{\rho}\right)}{\sum_{\ell=1}^{t+1} \exp\left(\frac{\langle y, v_\ell \rangle + (k-\ell)\gamma}{\rho}\right)}\right)$$

Since $c \geq \ln(1 + c)$, the following statement which we will show is in fact stronger.

$$n^{-\alpha} \geq \frac{k \max_{t+2 \leq \ell \leq j} \exp\left(\frac{\langle y, v_\ell \rangle + (k-t-2)\gamma}{\rho}\right)}{\exp\left(\frac{\langle y, v_{t+1} \rangle + (k-t-1)\gamma}{\rho}\right)}.$$

This can be rewritten as $-\rho\alpha \ln n \geq \rho \ln k + \max_{t+2 \leq \ell \leq j}\langle y, v_\ell \rangle - \langle y, v_{t+1} \rangle - \gamma$, or $\gamma \geq \rho(\ln k + \alpha \ln n) + \max_{t+2 \leq \ell \leq j}\langle y, v_\ell \rangle - \langle y, v_{t+1} \rangle$.

We know this last statement is true because the RHS is at most $\rho(1 + \alpha)\ln n + 20\sqrt{\frac{\ln n}{n}} + 2\beta$ which is smaller than $\gamma$, which is $40\sqrt{\frac{\ln n}{n}}$ (recall that $\beta = \gamma/\ln n$ and $\rho = \gamma/100\alpha \ln n$).

Since $E_x$ is true with probability $1 - 1/n^{10}$ Lemma 7, the lemma follows. $\qquad\square$

*Proof of Lemma 6.* Again, let $E_x$ be the event that $x$ satisfies $\forall i > t$ $|\langle v_i, x \rangle| \leq 10\sqrt{\frac{\ln n}{n}}$. Let us assume $E_x$ holds.

The value of $g(x)$ can be lower bound as follows. Since $\langle x, v_k \rangle \geq -10\sqrt{\frac{\ln n}{n}}$, $h(x) \geq f_k(x) \geq \rho \ln \exp\left(\frac{-10\sqrt{\frac{\ln n}{n}}}{\rho}\right) = -10\sqrt{\frac{\ln n}{n}}$. Since $h$ is 1-Lipschitz, $g(x) \geq -10\sqrt{\frac{\ln n}{n}} - 2\beta \geq -11\sqrt{\frac{\ln n}{n}}$ (because $\beta < 40/\sqrt{n \ln n}$).

For $x^* = \frac{1}{\sqrt{k}} \sum -v_i$, we know each $f_i(x^*)$ is at most $\rho \ln \left( k \exp \left( \frac{-1/\sqrt{k}+k\gamma}{\rho} \right) \right) + kn^{-\alpha}$. This is at most

$$\rho \ln k - \frac{1}{\sqrt{k}} + k\gamma + kn^{-\alpha}.$$

This in turn is at most $-0.8/\sqrt{k}$ since $k\gamma \leq 0.1/\sqrt{k}, n \geq \Omega(k^3), \alpha > 1$ and $\rho < 1/k$. So $g(x^*) \leq -0.8/\sqrt{k} + 2\beta < -0.7/\sqrt{k}$.

Since $\sqrt{\frac{\ln n}{n}} \ll 1/\sqrt{k}$, $g(x) > g(x^*) + 0.1/\sqrt{k}$ and so $x$ does not optimize $g$.

Since $E_x$ holds with probability $1 - 1/n^{10}$, the lemma follows. $\qquad\square$

## C  Parallel Randomized and Quantum Lower Bounds

Our randomized lower bound follows from two important properties satisfied by our hard class of functions. We abstract out these properties and define a generic class of hard functions. A class of functions $\mathscr{F}$ is an information-hiding class of functions if there is a sequence of 'partially-informed' functions that reveal very little new information, with $\mathscr{F}$ containing the 'fully-informed' functions.

An example the reader may want to keep in mind is the following 'Guess the numbers' problem. For a sequence of numbers $A = (a_1, \ldots, a_m) \in [N]^m$, consider the function $f_A$ that takes as input a sequence $B \in [N]^m$ and returns the sequence $A_{\leq i} 0^{m-i}$ where $i \in [m]$ is the maximum number such that $A_{<i} = B_{<i}$. The task is to learn $A$. An example 'partially-informed' function would be $f_{A_{\leq i} 0^{m-i}}$. This hides information in the sense that if one doesn't know $A_{\leq i}$ (say $A$ is chosen uniformly at random), then for most inputs the output of $f_A$ would be the same as the output of $f_{A_{\leq i} 0^{m-i}}$, and of course, for no input does the output of $f_{A_{\leq i} 0^{m-i}}$ reveal any more information than $A_{\leq i}$.

**Definition 6** (Information-Hiding Class of Functions). *Let $\mathcal{R} = (\mathcal{R}_1, \ldots, \mathcal{R}_m)$ be a random variable defining a sequence of functions $f_1, \ldots, f_m$ in the sense that setting a value of $\mathcal{R}_{\leq i}$ fixes the function $f_i$. The class of functions $\{f_m\}$ obtained by ranging over the various values of $\mathcal{R}$ is an $m$-step $(\delta_1, \delta_2)$-information-hiding class of functions (under the distribution $\mathcal{R}$) if the sequence satisfies the following properties.*

1. *For all $1 \leq i < m$ and any setting of $\mathcal{R}_{<i}$,*

$$\forall x \in \mathbb{R}^n : \Pr_{\mathcal{R}_{\geq i} | \mathcal{R}_{<i}} (O_{f_m}(x) = O_{f_i}(x)) \geq 1 - \delta_1$$

   *where $O_f(x)$ is the information about $f$ that the model allows us to query at $x$ (for example the function value, gradient and perhaps higher order derivatives if our queries provide them).*

2. *For any setting of $\mathcal{R}_{<m}$,*

$$\forall x \in \mathbb{R}^n : \Pr_{\mathcal{R}_m | \mathcal{R}_{<m}} (x \text{ is a correct output for } f_m) \leq \delta_2.$$

As a corollary of Lemmas 5 and 6, we see that our class of hard functions were indeed information-hiding functions.

**Corollary 1.** *Let $V = (v_1, \ldots, v_k)$ be the random variable that is distributed Haar randomly from the possible choices of $k$ orthonormal vectors from $\mathbb{R}^n$. The sequence of functions $g_1, \ldots, g_k$ is a $k$-step $(n^{-10}, n^{-10})$-information-hiding class of functions when the allowed queries are function values and derivatives up to the $p$th order derivative.*

We now prove the hardness of information-hiding classes of functions. We start with the setting of parallel randomized algorithms.

**Theorem 4.** *Let $\mathscr{F}$ be an $m$-step $(\delta_1, \delta_2)$-information-hiding class of functions under the distribution $\mathcal{R}$. Then for any parallel query algorithm $\mathcal{A}$ making $K$ queries per round and using less than $m$ rounds, the probability that the algorithm outputs a correct output for $f$ distributed according to $\mathcal{R}$ is at most $\delta_2 + mK\delta_1$.*

*Proof.* Let the success probability of $\mathcal{A}$ be $p_{\text{succ}}$ when $V$ is distributed Haar randomly. We can fix the randomness of $\mathcal{A}$ to get a deterministic algorithm $\mathcal{B}$ with success probability at least $p_{\text{succ}}$ on the same distribution.

Let us denote the transcript of $\mathcal{B}$ as $T = (S_1, S_2, \ldots, S_{m-1}, x_{\text{out}})$ where $S_i$ is the set of queries made in the $i$th round and $x_{\text{out}}$ is the output of the algorithm. Note that these are random variables that depend only on $\mathcal{R}$. We now create hybrid transcripts $T^{(i)}$ for $0 \leq i \leq m - 1$. The hybrid transcript $T^{(i)} = (S_1^{(i)}, \cdots, S_{m-1}^{(i)}, x_{\text{out}}^{(i)})$ is defined as the transcript of $\mathcal{B}$ when, for all $j \leq i$, the oracles calls in round $j$ (which are supposed to be to $\mathsf{Q}_{f_m}$) are replaced with oracle calls to $\mathsf{Q}_{f_j}$. Note that

- For any $V$, $T = T^{(0)}$.

- $T^{(m-1)}$ is a function of $\mathcal{R}_{\leq m-1}$.

- For any $V$, if the answers of $\mathsf{Q}_{f_m}$ on $S_i^{(i-1)}$ are the same as the answers of $\mathsf{Q}_{f_i}$ on $S_i^{(i)}$ then $T^{(i-1)} = T^{(i)}$. This is because they have queried the same oracles in their first $i-1$ calls, given the same inputs in the $i$th call and gotten the same output, and have been querying the same oracles thereafter.

We start with the observation that

$$\Pr_{\mathcal{R}}[x_{\text{out}}^{(m-1)} \text{ is } \epsilon\text{-optimal}] = \mathbb{E}_{\mathcal{R}_{<m}}\left[\Pr_{\mathcal{R}_m | \mathcal{R}_{<m}}[x_{\text{out}}^{(m-1)} \text{ is } \epsilon\text{-optimal}]\right] \tag{45}$$
$$\leq \delta_2. \qquad \text{(by property 2 in Definition 6)}$$

Next we show that $\Pr_{\mathcal{R}}[x_{\text{out}}^{(m-1)} = x_{\text{out}}] \geq 1 - mK\delta_1$ which will complete the proof.

$$\Pr_{\mathcal{R}}[x_{\text{out}} \neq x_{\text{out}}^{(m-1)}] \leq \sum_{i \in [m-1]} \Pr_{\mathcal{R}}[x_{\text{out}}^{(i-1)} \neq x_{\text{out}}^{i}] \tag{46}$$

$$\leq \sum_{i \in [m-1]} \Pr_{\mathcal{R}}[T^{(i-1)} \neq T^{(i)}] \tag{47}$$

$$\leq \sum_{i \in [m-1]} \Pr_{\mathcal{R}}[\mathsf{Q}_{f_m}(S_i^{(i-1)}) = \mathsf{Q}_{f_i}(S_i^{(i)})] \tag{48}$$

$$\leq \sum_{i \in [m-1]} \mathbb{E}_{\mathcal{R}_{<i}}\left[\Pr_{\mathcal{R}_{\geq i}|\mathcal{R}_{<i}}[\mathsf{Q}_{f_m}(S_i^{(i-1)}) = \mathsf{Q}_{f_i}(S_i^{(i)})]\right] \tag{49}$$

$$\leq mK\delta_1, \tag{50}$$

since $S_i^{(i-1)} = S_i^{(i)}$, and using property 1 in Definition 6 with a union bound over the inputs in each $S_i$. $\qquad\square$

We now turn to quantum query algorithms. In our quantum query model a $t$-query quantum query algorithm is a quantum circuit that uses a query oracle $t$ times. The query oracle is implemented by a unitary so that it supports queries in superposition. We allow arbitrarily high precision for the real numbers involved, and our lower bound is independent of the algorithm maker's choice of number of bits of precision. This is the same model used by and described in more detail in [GKNS21, Section 4.3].

We show that an information-hiding class of functions would be hard even for quantum query algorithms to compute. We can't use the above proof since we can't use a union bound on all queried points; a single quantum query may query exponentially many points in superposition. However, we know that a large fraction of this superposition is on points that don't reveal much information. The small fraction of points that do reveal information will not be noticeable to the quantum query algorithm since they are only a small fraction of the superpositioned points. We can then use the hybrid argument again to give a quantum query lower bound analogous to the classical one proved above.

**Theorem 5.** *Let $\mathscr{F}$ be an $m$-step $(\delta_1, \delta_2)$-information-hiding class of functions under the distribution $\mathcal{R}$. Then for any quantum query algorithm making less than $m$ queries, the probability that the algorithm outputs a correct output for $f$ distributed according to $\mathcal{R}$ is at most $\delta_2 + 4m\sqrt{\delta_1}$.*

The proof of this goes via what is commonly called the hybrid argument. Fix any quantum algorithm $A$ making at most $m-1$ queries, specified by the unitaries $U_{m-1}O_{f_m}U_{m-2}O_{f_m}\cdots U_1 O_{f_m}U_0$. Now we define a sequence of unitaries starting with $A_0 = A$ as follows:

$$
\begin{aligned}
A_0 &\coloneqq U_{m-1}O_{f_m}U_{m-2}O_{f_m}\cdots O_{f_m}U_1 O_{f_m}U_0 \\
A_1 &\coloneqq U_{m-1}O_{f_m}U_{m-2}O_{f_m}\cdots O_{f_m}U_1 O_{f_1}U_0 \\
A_2 &\coloneqq U_{m-1}O_{f_m}U_{m-2}O_{f_m}\cdots O_{f_2}U_1 O_{f_1}U_0 \\
&\vdots \\
A_{m-1} &\coloneqq U_{m-1}O_{f_{m-1}}U_{m-2}O_{f_{m-2}}\cdots O_{f_2}U_1 O_{f_1}U_0
\end{aligned}
\tag{51}
$$

Property 1 provides us with the following lemma.

**Lemma 8** ($A_t$ and $A_{t-1}$ have similar outputs). *Let $A$ be a $m-1$ query algorithm and let $A_t$ for $t \in [m-1]$ be the unitaries defined in eq. (51). Then*

$$
\mathop{\mathbb{E}}_{\mathcal{R}}\left(\|A_t|0\rangle - A_{t-1}|0\rangle\|^2\right) \le 4\delta_1.
\tag{52}
$$

*Proof.* From the definition of the unitaries in eq. (51) and the unitary invariance of the spectral norm, we see that $\|A_t|0\rangle - A_{t-1}|0\rangle\| = \|(O_{f_t} - O_{f_m})U_{t-1}O_{f_{t-1}}\cdots O_{f_1}U_0|0\rangle\|$. Let us prove the claim for any fixed choice of vectors $\mathcal{R}_{\le t-1}$, which will imply the claim for any distribution over those vectors. Once we have fixed these vectors, the state $U_{t-1}O_{f_{t-1}}\cdots O_{f_1}U_0|0\rangle$ is a fixed state, which we can call $|\psi\rangle$. Thus our problem reduces to showing for all quantum states $|\psi\rangle$,

$$
\mathop{\mathbb{E}}_{\mathcal{R}_{\ge t}|\mathcal{R}_{<t}}\left(\|(O_{f_t} - O_{f_m})|\psi\rangle\|^2\right) \le 4\delta_1.
\tag{53}
$$

Now we can write an arbitrary quantum state as $|\psi\rangle = \sum_x \alpha_x |x\rangle|\phi_x\rangle$, where $x$ is the query made to the oracle, and $\sum_x |\alpha_x|^2 = 1$. Thus the LHS of eq. (53) is equal to

$$
\mathop{\mathbb{E}}_{\mathcal{R}_{\ge t}|\mathcal{R}_{<t}}\left(\left\|\sum_x \alpha_x(O_{f_t} - O_{f_m})|x\rangle|\phi_x\rangle\right\|^2\right) \le \sum_x |\alpha_x|^2 \mathop{\mathbb{E}}_{\mathcal{R}_{\ge t}|\mathcal{R}_{<t}}\left(\|(O_{f_t} - O_{f_m})|x\rangle|\phi_x\rangle\|^2\right).
\tag{54}
$$

Since $|\alpha_x|^2$ defines a probability distribution over $x$, we can again upper bound the right hand side for any $x$ instead. Since $O_{f_t}$ and $O_{f_m}$ behave identically for some inputs $x$, the only nonzero terms are those where the oracles respond differently, which can only happen if $O_{f_t}(x) \ne O_{f_m}(x)$. When the response is different, we can upper bound $\|(O_{f_t} - O_{f_m})|x\rangle|\phi_x\rangle\|^2$ by 4 using the triangle inequality. Thus for any $x \in \mathbb{R}^n$, we have

$$
\mathop{\mathbb{E}}_{\mathcal{R}_{\ge t}|\mathcal{R}_{<t}}\left(\|(O_{f_t} - O_{f_m})|x\rangle|\phi_x\rangle\|^2\right) \le 4\mathop{\Pr}_{\mathcal{R}_{\ge t}|\mathcal{R}_{<t}}\left(O_{f_t}(x) \ne O_{f_m}(x)\right) \le 4\delta_1,
\tag{55}
$$

where the last inequality follows from Property 1. $\qquad\square$

And Property 2 provides us with the following.

**Lemma 9** ($A_{m-1}$ does not solve the problem). *Let $A$ be a $m-1$ query algorithm and let $A_{m-1}$ be defined as above. Let $p_R$ be the probability distribution over $x \in B(\vec{0}, 1)$ obtained by measuring the output state $A_{m-1}|0\rangle$ when the randomness $\mathcal{R}$ is fixed to $R$. Then $\Pr_{R\sim\mathcal{R}, x\sim p_R}(x$ is a correct output$) \le \delta_2$.*

*Proof.* Let us establish the claim for any fixed choice of $\mathcal{R}_{<m}$, since if the claim holds for any fixed choice of these vectors, then it also holds for any probability distribution over them. For a fixed choice of vectors, this claim is just $\Pr_{\mathcal{R}_m, x\sim p_R}(x$ is a correct output$) \le \delta_2$. Now since the algorithm $A_{m-1}$ only has oracles $O_{f_i}$ for $i < m$, the probability distribution $p_R$ only depends on $\mathcal{R}_{<m}$. Since these

are fixed, this is just a fixed distribution $p$. So we can instead establish our claim for all $x \in B(\vec{0}, 1)$, which will also establish it for any distribution.

So what we need to establish is that for any $x \in \mathbb{R}^n$, $\Pr_{\mathcal{R}_m}(x \text{ is a correct output}) \leq \delta_2$ which is what Property 2 gives us. $\qquad\square$

**Lemma 10** (*A does not solve the problem*). *Let $A$ be an $m-1$ query algorithm. Let $p_R$ be the probability distribution over $x \in B(\vec{0}, 1)$ obtained by measuring the output state $A|0\rangle$ when the randomness $\mathcal{R}$ is fixed to $R$. Then $\Pr_{R \sim \mathcal{R}, x \sim p_R}(x \text{ is a correct output}) \leq \delta_2 + 4m\sqrt{\delta_1}$.*

*Proof.* Let $P_R$ be the projection operator that projects a quantum state $|\psi\rangle$ onto the space spanned by vectors $|x\rangle$ for $x$ such that $x$ is a correct output when $\mathcal{R} = R$. Then $\|P_R A|0\rangle\|^2 = \Pr_{x \sim p_R}(x \text{ is a correct output})$. We know from Lemma 9 that $\mathbb{E}_{R \sim \mathcal{R}}(\|P_R A_{m-1}|0\rangle\|^2) \leq \delta_2$. We prove our upper bound on the probability by showing that it is approximately the same as $\mathbb{E}_{R \sim \mathcal{R}}(\|P_R A_{m-1}|0\rangle\|^2)$.

Lemma 8 states that for all $1 \leq t < m$, $\mathbb{E}_{\mathcal{R}}(\|A_t|0\rangle - A_{t-1}|0\rangle\|^2) \leq 4\delta_1$. Using telescoping sums and the Cauchy-Schwarz inequality, we see that

$$\mathbb{E}_{\mathcal{R}}(\|A_{m-1}|0\rangle - A|0\rangle\|^2) \leq \mathbb{E}_{\mathcal{R}}\left(\left(\sum_{t \in [m-1]} \|A_t|0\rangle - A_{t-1}|0\rangle\|\right)^2\right) \tag{56}$$

$$\leq \mathbb{E}_{\mathcal{R}}\left(\sum_{t \in [m-1]} \|A_t|0\rangle - A_{t-1}|0\rangle\|^2\right)\left(\sum_{t \in [m-1]} 1^2\right) \leq 4\delta_1 \cdot m \cdot m. \tag{57}$$

For all $R$, $\left|\|P_R A_{m-1}|0\rangle\| - \|P_R A|0\rangle\|\right| \leq \|P_R A_{m-1}|0\rangle - P_R A|0\rangle\| = \|P_R(A_{m-1}|0\rangle - A|0\rangle)\| \leq \|A_{m-1}|0\rangle - A|0\rangle\|$. Hence

$$\mathbb{E}_{R \sim \mathcal{R}}\left(\left(\|P_R A_{m-1}|0\rangle\| - \|P_R A|0\rangle\|\right)^2\right) \leq 4m^2 \delta_1. \tag{58}$$

We want an upper bound on $\mathbb{E}_{R \sim \mathcal{R}}(\|P_R A|0\rangle\|^2 - \|P_R A_{m-1}|0\rangle\|^2)$, which is no larger than $2\mathbb{E}_{R \sim \mathcal{R}}(\|P_R A|0\rangle\| - \|P_R A_{m-1}|0\rangle\|)$ since $\|P_R A|0\rangle\| + \|P_R A_{m-1}|0\rangle\| \leq 2$. We get such a bound by applying Jensen's inequality to eq. (58): $\mathbb{E}_{R \sim \mathcal{R}}(\|P_R A|0\rangle\| - \|P_R A_{m-1}|0\rangle\|) \leq 2m\sqrt{\delta_1}$, and so $\mathbb{E}_{R \sim \mathcal{R}}(\|P_R A|0\rangle\|^2 - \|P_R A_{m-1}|0\rangle\|^2) \leq 4m\sqrt{\delta_1}$.

We can now use linearity of expectation and upper bound our required probability as

$$\Pr_{R \sim \mathcal{R}, x \sim p_R}(x \text{ is a correct output}) = \mathbb{E}_{R \sim \mathcal{R}}(\|P_R A|0\rangle\|^2) \leq \delta_2 + 4m\sqrt{\delta_1}. \tag{59}$$
$\qquad\square$

The proofs of the quantum lower bound in Theorem 1 and the highly parallel lower bound alluded to after that now follow from Theorems 4 and 5 and Corollary 1.

**Corollary 2.** *The complexity of $\epsilon$-optimizing the class of functions $g$ is:*

- *$k$ rounds in the parallel randomized setting where in each round $K$ parallel queries are allowed, and $Kn^{-9} \ll 1$. (Note that by modifying the constants in the definition of the function, we can support $K$ being any polynomial in $n$.)*

- *$k$ queries in the quantum setting to get success probability larger than $n^{-4}$.*