# OpenReview forum: "Near-Optimal Lower Bounds For Convex Optimization For All Orders of Smoothness"
_NeurIPS.cc/2021/Conference — NeurIPS 2021 Spotlight_

### Official Review · Reviewer_2gRK · 2021-07-14

**Rating:** 6
**Confidence:** 3

**Summary:**

This paper stuides the complexity of optimizing highly smooth convex functions. For a positive integer p, one wants to find an $\epsilon$-approximate minimum of a convex function f, given oracle aceess to the function and its first p derivatives, assuming that the p-th derivative of f is Lipschitz. The near-optimal algorithms exist in deterministic setting while it remains a gap between the upper bounds and lower bounds for randomized algorithms. The contribution of this paper is to provide a new lower bound that matches the best existing upper bound (up to log factors), and shows that it holds not only for randomized algorithms, but also quantum algorithms.

To prove the desired lower bound, the paper started with a classical nonsmooth lower bound instance in Eq. (8) and employed a new random smoothing technique. By this construction, the proof strategy for nonsmooth convex optimization can be executed on the smoothed instance considered in this paper, thereby giving us a lower bound for p-th order smooth convex optimization. Using this idea, the authors prove the lower bound for the desired class of optimization problems.

**Limitations And Societal Impact:**

Justify why studying the randomized high-order optimization algorithms and quantum algorithms is important given prior works.

**Main Review:**

Pros: Overall, I think the results are very interesting. Similar ideas have been widely used for proving complexity lower bound in the recent papers of [GN15] and [DG20] when p=1.

[GN15] C. Guzman and A. Nemirovski "On Lower Complexity Bounds for Large-Scale Smooth Convex Optimization", Journal of Complexty, 31 (1): 1-14, 2015.

[DG20] J. Diakonikolas and C. Guzmán. "Lower Bounds for Parallel and Randomized Convex Optimization", JMLR, 21 (5): 1-31, 2020.

The novelty of this paper appears to be a different smoothing operator which extends the previous proof strategy for proving query lower bounds for high-order smooth convex optimization. Indeed, the inf-conv smoothing plays a fundamental role when p=1 and it is difficult to generalize it to p>=2. The authors did a great job in explaining the intuition behind their construction and I believe all the arguments are technically sound. I can see some new elements, but my knowledge of the previous work in this area is not deep enough to evaluate technical novelty very well.

Cons: Despite the tight lower bound, the motivation of this work is elusive. Indeed, the high-order randomized algorithm seems vary rare in the community and the only possible one I am aware of is the following article.

V. S. Amaral, R. Andreani, E. G. Birgin, D. S. Marcondes, J. M. Martínez, "On complexity and convergence of high-order coordinate descent algorithms". (https://arxiv.org/abs/2009.01811)

There are also some other near-optimal high-order algorithms in the literature:

T. Lin and M. I. Jordan. "A control-theoretic perspective on optimal high-order optimization". (https://arxiv.org/abs/1912.07168)

M. Marques Alves. "Variants of the A-HPE and large-step A-HPE algorithms for strongly convex problems with applications to accelerated high-order tensor methods". (https://arxiv.org/abs/2102.02045)

Nevertheless, the high-order randomized optimization algorithms may have the potentials for structured nonconvex optimization since they may return better stationary points and thus have practical implications in machine learning.

P. Xu, F. Roosta-Khorasani, M. W. Mahoney. "Second-order optimization for non-convex machine learning: an empirical study". ICDM 2020.

But a lower bound proved for the convex case in this paper seems to be stretching this a little far.

Moreover, I have some question about the extent to which this work is in scope for NeurIPS.  Indeed, the randomzied high-order optimization algorithms have received less attention and seem less competitive with the randomized first-order optimization algorithms for convex cases. Therefore, this paper seems like a somewhat good contribution that however would be of interest to a smallish subset of NeurIPS attendees.

**Time Spent Reviewing:**

4 hours

---

> ### Author Response · Authors · 2021-08-09
> **Thank you!**
>
> Thanks for your comments. The motivation for our work was two-fold. First, it is true that in convex optimization in the setting that we consider (with only black box access to oracles), randomization offers no benefit in oracle queries (as our result shows). But there are related settings, such as the setting where the objective function is available as a finite sum (so it is not black box anymore), where SVRG, a randomized algorithm, offers a speedup over all known deterministic algorithms. Given that the known lower bounds for randomized algorithms for our problem were conspicuously different from the complexity of the best known algorithms, one might think that perhaps randomization could help in this situation? In any case, it was not obvious to us that randomization cannot help over deterministic algorithms, so we tried to prove it formally. Second, when it comes to quantum algorithms, all bets are off. Quantum algorithms provide mysterious speedups in many situations and we have a very incomplete understanding of when they provide a speedup. Even within convex optimization, quantum algorithms offer surprising speedups. It is known that quantum algorithms can compute a gradient query from a function oracle query with constant uses of the oracle (whereas non-quantum algorithms need linear-in-the-dimension queries). Quantum algorithms are also known to give a speedup for the problem of computing the volume of a convex body. Given such examples, it was very unclear whether quantum algorithms could offer any speedup for this problem, and hence we set out to understand the problem's quantum complexity.
>
> We also agree with you that higher order methods are not commonly used, but note that our lower bound is new even for p=2 (where we assume access to the Hessian). Starting from the seminal work of [NP06], there have been many works in optimization which assume access to a Hessian oracle, so we feel this isn't a very obscure setting. We agree that the general order-p methods may not be practical, but it is still nice to know they are optimal and further improvement is not expected.
>
> [NP06] Cubic regularization of Newton method and its global performance, by Y. Nesterov and B. Polyak, Math. Programming 108

---

### Official Review · Reviewer_wfjt · 2021-07-16

**Rating:** 9
**Confidence:** 4

**Summary:**

The paper provides nearly tight lower bounds on the complexity of highly smooth convex optimization, using an oracle model where the oracle gives access to the function and its first p derivatives. The lower bounds extend previous bounds on deterministic algorithms to the randomized and quantum setting.

**Limitations And Societal Impact:**

Yes. No negative societal impact is applicable as this is purely a theoretical paper.

**Main Review:**

This is an excellent paper that resolves a relevant open question in the field of convex optimization. In addition, the quantum lower bound seems quite novel.

Questions:

Do the authors think that the logarithmic factor in their bound could be improved with a tighter analysis and/or slight modification of the approach, or is it more likely to be fundamental and require different techniques to improve (or even, might it actually be tight)?

Could these methods be generalized to obtain improved lower bounds for broader classes of functions, such as highly smooth star-convex functions (see https://arxiv.org/abs/1906.11985)?

Can reduction be applied to generalize these lower bound results to the constrained setting (i.e., where the domain is a closed convex set that may not necessarily be a ball)? If the results can be generalized to the constrained setting, this might be a nice note to add. If not, what are the key challenges?

**Time Spent Reviewing:**

4

---

> ### Author Response · Authors · 2021-08-09
> **Thank you!**
>
> Thanks for your review and comments. Here are some answers to your questions:
>
> - We believe the logarithmic factor is not needed, but it might need some modification of the function class/proof strategy to show that. For example, while our results have log factors even for p = 1, earlier results of [GN15] and [DG20] show that the log factors are not necessary for p = 1.
> - The deterministic lower bounds for quasar convex functions in [HSS19] directly work with a smooth function. In order to apply the current approach, we might first need to prove a lower bound for **nonsmooth** quasar convex functions, where the nonsmoothness is generated through a 'max' operation, and then perhaps use smoothing to obtain lower bounds for smooth functions. This is certainly an interesting direction to pursue.
> - We may not expect lower bounds for any given arbitrary constraint set e.g., if the constraint set is a singleton, there is nothing to optimize. We may need to formulate the question more carefully (to avoid such trivial issues) to investigate this further.

---

> > ### Comment · Reviewer_wfjt · 2021-09-03
> > **Thank you for the reply**
> >
> > After the response phase, my opinion remains that this is a strong paper clearly deserving of acceptance.
> >
> > Minor note: In the abstract, "PLMR" should be "PMLR".

---

### Official Review · Reviewer_jZkZ · 2021-07-16

**Rating:** 7
**Confidence:** 3

**Summary:**

This paper studies the complexity of optimizing convex functions with Lipschitz $p$-th derivative. The authors prove a new lower bound $\tilde{O} (1 / \epsilon^{ \frac{2}{3p+1} } )$ that holds for deterministic, randomized and quantum algorithms. This lower bound matches the upper bound up to log factors.

**Limitations And Societal Impact:**

yes

**Main Review:**

[strength]

  The construction of the hard functions is different from previous works. The authors use a novel combination of softmax function and randomized smoothing. The claims are well supported by theoretical analysis. The method is applicable to a general function class, information-hiding class of functions.

  [weakness]

 The paper is not very well organized. Section 2 is a bit lengthy. while the detailed proof is a little brief. More interpretation of Lemmas 5 and 6 are preferred.

  [typos]

  Equation 28: a pair of parentheses is missing.

  Equations 30 and 31: the sign $=$ should be the sign $\neq$.


**Time Spent Reviewing:**

20

---

> ### Author Response · Authors · 2021-08-09
> **Thank you!**
>
> Thanks for the suggestion. We do have a little more space remaining so we will add some intuition for Lemmas 5 and 6.

---

> > ### Comment · Reviewer_jZkZ · 2021-08-31
> > **Thanks for the response**
> >
> > I have read the authors’ rebuttals and other reviewers' comments, and keep my score.

---

### Decision · Program_Chairs · 2021-09-27

**Decision:**

Accept (Spotlight)

**Comment:**

Thank you for submitting your paper to NeurIPS'21.

Despite some minor concerns that should be addressed in the camera-ready version, the reviewers agreed that the paper is worthy of acceptance and is presented at NeurIPS'21. Congratulations!